# A multivariate decomposition analysis of drivers of overweight and obesity among Ghanaian women

Joseph Prince Mensah [1] ✉, Robert Akparibo [1], Afua Atuobi-Yeboah [2], Emmanuel Anaba[2], Laura Ann Gray [1], Isaac Boadu[2], Maxwell Bisala Konlan[2] & Richmond Aryeetey[2]

## Abstract

**Background** Overweight and obesity are rising globally, with Ghana experiencing significant increases among women over the past two decades, raising public health concerns. This study aimed to identify and quantify the key drivers of overweight and obesity among women of reproductive age in Ghana, analysing how these factors have contributed to prevalence changes over time. **Methods** Data from the 2003, 2008, 2014, and 2022 Ghana Demographic and Health Surveys were analysed using binary logistic regression to assess associations with factors such as age, wealth, and education. Multivariate decomposition analysis quantified the contributions of these factors to the observed increases in overweight and obesity prevalence over time. **Results** Here we show overweight and obesity among Ghanaian women rise significantly, reaching 43% in 2022. Key drivers of change in overweight and obesity include wealth, education, urban residence, age, and region. Women in the wealthiest quintile have three times the odds of overweight (aOR: 3.07 [2.02–4.67]) and over six times the odds of obesity (aOR: 6.73 [3.80–11.91]) compared to the poorest quintile. Decomposition analysis shows that 22.5% of the increase in prevalence was due to changes in population characteristics, such as marital and educational status. **Conclusions** Our findings reveal that socio-demographic changes in society, beyond individual behavioural factors, drive the rising overweight and obesity prevalence among Ghanaian women of childbearing age. These findings highlight the dynamic factors influencing weight outcomes and the need for tailored strategies addressing the diverse and evolving determinants of overweight and obesity in Ghanaian women.

## Plain language summary

Overweight and obesity prevalence are rising globally, posing health challenges. We examined the main factors influencing the changing rate of overweight and obesity among Ghanaian women. We used data from surveys undertaken in 2003, 2008, 2014, and 2022. We assessed the association between overweight and obesity and factors such as age, wealth, and education, as well as their contributions to the changes in overweight and obesity. Results show 43% of Ghanaian women were living with overweight or obesity in 2022. Key drivers included socio-demographic factors such as age and wealth. These results suggest that addressing overweight and obesity in Ghana requires tailored interventions targeting these drivers of overweight and obesity to mitigate their impact and improve population health outcomes.

Overweight and obesity are non-communicable diseases (NCDs) and are significantly associated with numerous other NCDs and disabilities, severely affecting quality of life and health worldwide[1,2]. Obesity is strongly associated with cardiovascular diseases, such as heart disease and stroke, type 2 diabetes, certain cancers (e.g., breast and colorectal), and respiratory disorders like sleep apnoea[3,4]. Beyond their health impacts, overweight and obesity adversely impact social and economic productivity, on a global scale[5].

The World Health Organization (WHO) reports a consistent rise in obesity over the past three decades, with one in eight individuals currently living with the condition globally[6]. As of 2022, the global prevalence of overweight among adults stood at 43%, with similar rates for men (43%) and women (44%), while 16% of adults were living with obesity[7]. Notably, there are significant disparities in overweight and obesity prevalence across different regions of the world[8]. Across Africa more broadly, pooled analyses (1990–2022) reveal a steady increase in mean body mass index and obesity prevalence among adults, with similar rising patterns among children and adolescents[9,10]. More specifically, in Sub Saharan Africa, estimates suggest that between 23% and 38% of adults living in urban areas have obesity or overweight[11]. These shifts are largely driven by rapid urbanisation, nutrition

[1]School of Medicine and Population Health, University of Sheffield, Sheffield, UK. [2]School of Public Health, University of Ghana, Legon, Accra, Ghana. ✉e-mail: jpmensah1@sheffield.ac.uk

transitions toward energy-dense processed foods, and declining physical activity[12]. Furthermore, urban studies document shifts away from traditional staples toward more processed foods; ultra-processed food consumption is consistently linked with higher obesity odds[13]. A recent UNICEF and global monitoring reports now show that overweight and obesity in school-age children and adolescents outpace underweight in many countries, underscoring the urgency of intervention[14]. Together, these findings highlight the escalating public health challenge posed by overweight and obesity across Africa, emphasising the need for comprehensive, multi-sectoral strategies to address the shared behavioural and structural drivers.

Ghana has experienced a marked increase in overweight and obesity prevalence, particularly among women, raising public health concerns[15,16]. The 2022 Ghana Demographic and Health Survey (GDHS) reported that 43% of women aged 15–49 years were living with overweight or obesity, with the prevalence rising to 50% among women aged 20–49 years. This represents a marked increase from 30% in 2008 and 40% in 2014 among women aged 15–49 years[17]. Women in urban areas particularly have been affected, with prevalence rates significantly outpacing those in rural settings[18]. This growing public health challenge demands an urgent, coordinated, and multisectoral response, particularly as the economic and health burdens of obesity escalate nationally. The rising prevalence of obesity impacts individual health and burdens national health systems due to increased healthcare costs and demand for medical services, and therefore poses a substantial threat to the nation's overall health and productivity[5,19].

Some Ghanaian studies attributes the rising prevalence of overweight and obesity in Ghana to factors including urbanisation, shifts in dietary patterns and lifestyle choices, physical inactivity, and biological predispositions, especially among women[16,20,21]. The observed trend in overweight and obesity rates among Ghanaian women presents an important opportunity to examine the context-specific determinants and drivers behind this increase. Understanding these factors is crucial for informing policymakers and guiding effective interventions to tackle this public health challenge and support national development[22]. While obesity has been widely examined in high-income settings, there is limited evidence on how sociodemographic and economic factors shape obesity risk in sub-Saharan Africa, particularly Ghana. Existing studies have often described prevalence patterns but have not disentangled how disparities by age, education, marital status, wealth, and residence contribute to rising obesity trends[16], while others have examined the association between biological, behavioural and socio-demographic factors and overweight and obesity among women in Ghana[23,24]. This study addresses this gap by applying decomposition analysis to examine both the distribution of these characteristics and their changing influence on overweight and obesity among Ghanaian women of reproductive age.

It has previously been suggested that without an urgent public health intervention, there could be further rises in the prevalence of overweight and obesity, as well as its associated health outcomes in the coming years in Ghana[23]. However, gaps remain in understanding how these drivers have contributed to the observed changes in prevalence over time. In order to identify key targets for intervention, it is necessary to understand the contributory factors and quantify their effect on the rise in prevalence over the years.

The aim of the study, therefore, was to identify and quantify the key drivers of overweight and obesity among women of reproductive age in Ghana over the past two decades. Using regression and decomposition analyses, we analyse data from the 2002, 2008, 2014, and 2022 GDHS to determine the contributions of specific factors to the observed prevalence. We find that socio-demographic changes in society, beyond individual behavioural factors, are driving an increase in overweight and obesity prevalence among Ghanaian women of childbearing age.

## Methods
### Data source
This study utilised nationally representative data from the GDHS conducted in 2003, 2008, 2014, and 2022. These surveys are part of the global Demographic and Health Surveys (DHS) programme, which collects standardised and comparable data on key demographic and health indicators across multiple countries. The primary purpose of the DHS programme is to provide high-quality data to inform policy formulation, evaluate health programmes, and monitor trends in population and health outcomes over time.

The GDHS datasets are publicly available through the DHS program website (https://dhsprogram.com) and we obtained all requisite permissions to access and use these data. The datasets analysed in this study include detailed information on demographic, socioeconomic, and health-related variables at both regional and urban/rural levels, which is critical for examining trends in obesity and related determinants within households and across the general population. It facilitates the decomposition of changes in overweight and obesity prevalence among women of reproductive age over the years spanned by these surveys. The inclusion of regional and stratified data also enables robust sub-national analyses ensuring insights into urban/rural disparities and regional differences in obesity trends.

### Sampling design
The GDHS employs a stratified, two-stage cluster sampling design to ensure representativeness at both national and sub-national levels. In the first stage, clusters or enumeration areas are selected based on probability proportional to size, considering both urban and rural locations within each administrative region of Ghana. The second stage involves the selection of households within these clusters, refined through detailed household listings. The survey team obtained ethics approval from the Ethical Review Committee of the Ghana Health Service, and international best practices in research ethics, such as obtaining informed consent to participate, were observed throughout the survey processes.

### Outcome measures: overweight and obesity
This study focused on two mutually exclusive primary health outcomes: overweight and obesity among women aged 15–49 years. Overweight was defined as a body mass index (BMI) $\geq 25 \text{ kg/m}^2$ and $<30 \text{ kg/m}^2$, while obesity was defined as a BMI of $30 \text{ kg/m}^2$ or higher. BMI was calculated using the formula: weight in kilograms divided by height in meters squared $(\text{kg/m}^2)$. Data on weight and height were measured objectively by biomarker technicians of the GDHS, ensuring accuracy and consistency. Women who were pregnant at the time of the survey or who had given birth within the two months preceding the survey were excluded from the analysis. This exclusion criteria ensures that the reported BMI accurately reflects non-pregnancy-related overweight or obesity status.

### Explanatory variables
The selection of explanatory variables for this study was guided by a comprehensive review of the literature on the determinants and drivers of overweight and obesity in adult populations, globally, and in similar low- and middle-income settings[25,26]. The chosen variables capture a wide range of individual and household characteristics.

Household-level variables:
- Place of residence: urban or rural classification.
- Region: one of Ghana's (formerly 10) administrative regions.
- Number of household members: a measure of household size.
- Wealth index: a measure of household economic status derived from asset ownership and housing characteristics.
- Sex of household head: whether the person who makes decisions for the household is male or female.

Individual-level variables:
- Sociodemographic characteristics: age, marital status, education level, religion, ethnicity, employment status, and occupation.
- Behavioural, lifestyle and reproductive factors: smoking status, occupation, employment status, contraceptive use, number of births in the last five years, and media exposure (frequency of television viewing, radio listening, and newspaper reading).

## Statistical analysis

Data analyses were conducted using cases with complete BMI data. Descriptive statistics were generated to summarise the distribution of overweight and obesity across survey years. Percentages were used to report the prevalence of overweight and obesity. Chi-squared tests were employed to evaluate associations between categorical variables and weight status.

To assess the relationship between explanatory variables and overweight or obesity, survey-weighted binary logistic regression models were fitted using data from the 2022 survey. This method accounts for the complex survey design, including stratification and clustering, and applies sampling weights to improve generalisability. Variance inflation factors (VIFs) were computed to assess multicollinearity among explanatory variables. Ethnicity was excluded from the final models due to collinearity with the region. Smoking status, although relevant to weight outcomes[27], was excluded because of the small proportion of smokers in the sample (1%), which limited statistical power. Separate logistic regression models were fitted for overweight and obesity, estimating the adjusted odds ratios (aORs) for each outcome relative to healthy weight status.

To test the robustness of findings across the years, a pooled logistic regression analysis was conducted using data from all survey years (2003–2022). A dummy variable for survey year was added to adjust for temporal effects. Survey sample weights were de-normalised for the pooled regression by dividing each sample weight by the women's survey sampling fraction for each year, reflecting the proportion of women aged 15–49 years interviewed relative to the total population of women in that age group, as estimated by OurWorldinData (ourworldindata.org/population-growth). Model fit was assessed using the Nagelkerke pseudo-$R^2$ statistic.

To explore factors driving changes in overweight and obesity prevalence over time, the Blinder–Oaxaca decomposition method for non-linear models was applied[28]. This method has previously been used to analyse the changes in infant mortality rates and anaemia reduction using data from the GDHS[29,30], but has not been applied to decompose the trend in overweight and obesity prevalence in Ghana. This approach decomposes differences in overweight and obesity prevalence between a time period into three components:

1. Endowment effects: difference due to change in population characteristics (e.g., proportion of women living in rural areas),
2. Coefficient effects: difference due to effects of these characteristics on the outcome (e.g, the impact of wealth on obesity), and
3. Interaction effects: difference due to the interaction between change in population characteristics and their coefficient effect.

Separate decomposition analyses were conducted to analyse the changes in overweight and obesity prevalence between 2022 and 2014, as well as between 2022 and 2003. The latter was done as a sensitivity analysis to assess how the key drivers contributed to the observed prevalence trend over the longer time period. Variables significantly associated with overweight or obesity in the 2022 regression model were included in the decomposition models.

Descriptive and regression analyses were conducted in R (version 4.4.1) using custom scripts developed for this study[31], while decomposition analyses were performed in Stata 18 (Stata Corporation, College Station, TX) due to its support for non-linear decomposition methods suitable for binary outcomes. All analyses accounted for the complex survey design by applying sampling weights and adjusting for clustering using the 'svy' command in Stata and the 'survey' package in R to obtain robust standard errors. Statistical significance was determined at a $p$-value < 0.05 for all tests.

## Results

### Summary of population characteristics

The demographic and socioeconomic characteristics of the study population across the four GDHS survey years (2003, 2008, 2014, and 2022) are summarised in Supplementary Data 1, with all estimates reported as weighted prevalence. After excluding 14,621 participants due to missing anthropometric data, pregnancy, or recent childbirth (Table 1), the analysis includes complete data from 20,396 women of reproductive age. In 2022, the mean age among women aged 15–49 was 29.9 years, with 46% residing in female-headed households. The proportion of women married or cohabiting was 51.9%.

Educational attainment shows marked improvement over time; the proportion of women with no education dropped from 18.9% in 2014 to 15.5% in 2022, while those with secondary or higher education increased from 52.8% in 2003 to 70.6% in 2022. Urban residency among the study population also increased, rising from 49.2% in 2003 to 58.1% in 2022 (Supplementary Data 1).

### Prevalence of overweight and obesity

The prevalence of overweight and obesity demonstrates a consistent upward trend, as shown in Supplementary Data 2. Between 2003 and 2022, the mean BMI of women aged 15–49 years rose from 23.10 kg/m² to 25.04 kg/m². In 2003, 25.3% of women were classified as having overweight or obesity. This percentage rose to 29.9% in 2008, 40.1% in 2014, and 43.0% in 2022. This represents a 17.7 percentage point increase over nearly two decades. The rising prevalence of overweight and obesity is not uniform across all demographic groups. Women living in urban areas, those in wealthier households, and those with higher educational attainment were disproportionately affected (Supplementary Data 2).

### Drivers of overweight and obesity

The results of the 2022 GDHS and 2003–2022 pooled binary logistic regression analyses, identifying key individual- and household-level factors associated with overweight and obesity, are presented in Supplementary Data 3. These factors included household-level characteristics such as place of residence, region, household size, wealth index, and sex of the household head, and individual-level demographic factors such as age, marital status, education, contraceptive use, number of births, and frequency of television watching.

Age is a significant driver of overweight and obesity across all survey years. Each additional year of age was associated with higher odds of overweight or obesity. Specifically, the aOR for overweight of those in higher age categories compared to lower ages was 1.05 (95% CI: 1.04–1.07, $p < 0.001$), while the aOR for obesity was 1.09 (95% CI: 1.08–1.10, $p < 0.001$). Marital status also played a significant role in weight outcomes, but its influence evolved over time. In 2003, women who had never been married had much lower odds of overweight compared to their married counterparts (aOR: 0.35, 95% CI: 0.25–0.49, $p < 0.001$) (Supplementary Data 4). By 2022, this protective effect had diminished, although the odds were still lower compared to women who were married or cohabiting (aOR: 0.67, 95% CI: 0.51–0.89, $p < 0.01$). However, this association did not persist among the rural population when compared to the urban population (urban aOR: 0.59, 95% CI: 0.41–0.86, $p < 0.01$; rural aOR: 0.85, 95% CI: 0.56–1.31, $p > 0.05$) (Supplementary Data 5).

Educational attainment is positively associated with overweight and obesity. In 2022, women with secondary or higher education were more likely to have obesity compared to those with no formal education (secondary education aOR: 1.73, 95% CI: 1.18–2.54, $p < 0.01$; higher education aOR: 1.87, 95% CI: 1.14–3.06, $p < 0.05$). This association remained consistent across both rural and urban populations (Supplementary Data 5). This result was consistent across the survey years (secondary education pooled aOR: 1.40, 95% CI: 1.13–1.73, $p < 0.01$; higher education pooled aOR: 1.48, 95% CI: 1.08–2.03, $p < 0.05$). The wealth index also emerges as a consistent and key driver of overweight and obesity. In 2022, women in the wealthiest quintile had more than three times the odds of overweight (aOR: 3.07, 95% CI: 2.02–4.67, $p < 0.001$) and over six times the odds of obesity (aOR: 6.73, 95% CI: 3.80–11.91, $p < 0.001$) compared to those in the poorest quintile. This association was evident for overweight among women in rural areas but was not statistically significant among women in urban areas (urban aOR: 1.95, 95% CI: 0.96–3.97, $p > 0.05$; rural aOR: 3.69, 95% CI: 1.96–6.95, $p < 0.001$) (Supplementary Data 5).

**Table 1 | Summary of Ghana Demographic and Health Surveys included in this study's analysis**

| Year | Number of households interviewed | Number of women aged 15–49 years interviewed | Eligible women's response rate (%) | Number of respondents with missing anthropometric data | Number of participants pregnant at time of survey | Number of participants who gave birth to child in the two months prior to the survey |
|---|---|---|---|---|---|---|
| 2003 | 6251 | 5691 | 95.7 | 345 | 435 | 102 |
| 2008 | 11,778 | 4916 | 96.5 | 102 | 365 | 89 |
| 2014 | 11,835 | 9396 | 97.3 | 4646 | 679 | 138 |
| 2022 | 17,933 | 15,014 | 98.0 | 7615 | 1111 | 347 |

Urban residency is also one of the significant drivers of overweight and obesity. By 2022, urban women had 64% greater odds of having obesity (aOR: 1.64, 95% CI: 1.27–2.12, $p < 0.001$) compared to rural women. Moreover, the odds of obesity among women in the Greater Accra region was higher compared to women in the Ashanti region, the two largest regions in Ghana by population (pooled aOR: 1.57, 95% CI: 1.28–1.94, $p < 0.001$). The opposite trend was observed in the Upper West region, where the aOR in the pooled regression was 0.46 (95% CI: 0.31–0.67, $p < 0.001$).

The pooled regression model revealed that women living in female-headed households have higher odds of overweight (aOR: 1.17, 95% CI: 1.05–1.31, $p < 0.01$), though this association was not significant in the 2022 regression model (aOR: 1.18, 95% CI: 0.96–1.44, $p > 0.05$). In 2022 however, each additional childbirth increased the odds of overweight by 16% (aOR: 1.16, 95% CI: 1.01–1.32, $p < 0.05$), while an increase in the number of household members reduced the odds of overweight by ~4% (aOR: 0.96, 95% CI: 0.93–1.00, $p < 0.05$). Additionally, modern contraceptive use is significantly associated with increased odds of overweight, with an aOR of 1.36 (95% CI: 1.12–1.65, $p < 0.01$). Among rural women, this association was observed with traditional contraceptive methods, whereas in urban areas, it was linked to modern methods (Supplementary Data 5).

**Multivariate decomposition analysis**

The multivariate decomposition analyses (Tables 2 and 3) quantify the contributions of changes in individual and household characteristics to the rising prevalence of overweight and obesity among Ghanaian women. Between 2014 and 2022, the prevalence of overweight and obesity increased by 3 percentage points, reaching 43% in 2022. This increase was primarily driven by coefficient effects (113.3% increase; $p < 0.01$), which reflects the difference in prevalence due to the evolving influence or impact of these characteristics. Overall shifts in the population characteristics alone (endowment effect) were not statistically significant (22.5% increase; $p = 0.318$), nor were changes in the interaction effect (35.8% decrease; $p = 0.066$).

In terms of the coefficient effect, only wealth (richest quintile: 2.42% decrease, $p < 0.05$), urban residence (6.06% increase, $p < 0.001$), and region (Greater Accra: 1.41% decrease, $p < 0.05$) made statistically significant contributions to the change in prevalence (Table 2). This suggests that urban residency and wealth continue to exert significant influence, with changes in their coefficient effects contributing to the observed change in overweight and obesity prevalence. For instance, the 2.42% negative contribution to prevalence from the coefficient effect of the richest quintile suggests that behavioural changes or conditions specific to women in this wealth group have reduced the impact of wealth on weight gain.

Furthermore, marital status and contraceptive use emerges as significant drivers of the prevalence trend within the endowment effect (Table 2). The rise in the proportion of women who have never married had a negative contribution to the increase in the overweight and obesity prevalence by 0.156 percentage points ($p < 0.05$). Conversely, the increased use of modern contraceptives in the 2022 population contributed 0.264 percentage points ($p < 0.05$) to the rise in overweight and obesity prevalence between 2014 and 2022. Other variables, such as age, urban residence, and wealth, do not show significant impacts within the endowment component.

Over the longer period from 2003 to 2022, the prevalence of overweight and obesity increases substantially, rising from 25.3% to 43.0% among women aged 15–49 years (Supplementary Data 2). The decomposition analysis for this period (Table 3) provides insight into the evolving impact of the drivers of overweight and obesity. Changes in population characteristics (endowment effect) account for a moderate but significant portion of the overall rise in overweight and obesity prevalence (22.5% increase, $p < 0.001$) (Table 3). The increased average age of the population contributed 0.94 percentage points ($p < 0.01$) to the rise in prevalence in this period.

Education is also a significant contributor to the endowment effect (Table 3), highlighting its influence as a socioeconomic factor to the rising trend in overweight prevalence. The findings show that an increase in the

**Table 2 | Multivariate decomposition of changes in overweight and obesity prevalence between Ghana DHS 2014 and 2022**

| Variable | 2014–2022 overweight and obesity prevalence difference due to | | |
|---|---|---|---|
| | Coefficients | Endowments | Interaction |
| Age (years) | 1.89 | $-9.28 \times 10^{-2}$ | $7.55 \times 10^{-3}$ |
| **Marital status** | | | |
| Married/Cohabiting | – | – | – |
| Never married | 1.61 | $-1.56 \times 10^{-1}$* | $-1.67 \times 10^{-1}$ |
| Previously married | $2.08 \times 10^{-1}$ | $-7.87 \times 10^{-3}$ | $7.32 \times 10^{-3}$ |
| **Educational attainment** | | | |
| No education | – | – | – |
| Primary | $5.52 \times 10^{-1}$ | $-3.22 \times 10^{-1}$* | $2.15 \times 10^{-1}$ |
| Secondary | $7.74 \times 10^{-1}$ | $2.39 \times 10^{-1}$ | $-5.83 \times 10^{-2}$ |
| Higher | $-1.11 \times 10^{-1}$ | $2.44 \times 10^{-1}$ | $5.66 \times 10^{-2}$ |
| **Wealth quintile** | | | |
| Poorest | – | – | – |
| Poorer | $-6.52 \times 10^{-2}$ | $3.84 \times 10^{-2}$ | $2.99 \times 10^{-3}$ |
| Middle | $-1.56$ | $-1.82 \times 10^{-1}$ | $-1.79 \times 10^{-2}$ |
| Richer | $-2.36$* | $1.40 \times 10^{-1}$ | $1.23 \times 10^{-1}$ |
| Richest | $-2.42$* | $-5.30 \times 10^{-1}$ | $-3.72 \times 10^{-2}$ |
| **Residence location** | | | |
| Rural | – | – | – |
| Urban | 6.06*** | $1.44 \times 10^{-1}$ | $-4.24 \times 10^{-1}$* |
| **Region** | | | |
| Ashanti | – | – | – |
| Brong−Ahafo | $-4.65 \times 10^{-1}$ | $-6.76 \times 10^{-2}$ | $1.14 \times 10^{-1}$ |
| Central | $-8.47 \times 10^{-2}$ | $-4.73 \times 10^{-2}$ | $1.76 \times 10^{-2}$ |
| Eastern | $2.06 \times 10^{-2}$ | $1.23 \times 10^{-2}$ | $1.78 \times 10^{-3}$ |
| Greater Accra | $-1.41$* | $1.66 \times 10^{-1}$ | $-4.29 \times 10^{-1}$ |
| Northern | $9.72 \times 10^{-2}$ | $-3.44 \times 10^{-2}$ | $-5.64 \times 10^{-3}$ |
| Upper East | $-5.79 \times 10^{-1}$* | $-1.77 \times 10^{-1}$* | $2.60 \times 10^{-1}$ |
| Upper West | $-2.47 \times 10^{-1}$ | $-2.94 \times 10^{-2}$ | $3.53 \times 10^{-2}$ |
| Volta | $-1.39 \times 10^{-1}$ | $3.51 \times 10^{-2}$ | $-1.73 \times 10^{-2}$ |
| Western | $-4.15 \times 10^{-1}$ | $9.81 \times 10^{-2}$ | $-1.67 \times 10^{-1}$ |
| **TV watching frequency** | | | |
| Not at all | – | – | – |
| At least once a week | $-5.43 \times 10^{-1}$ | $4.44 \times 10^{-1}$ | $1.17 \times 10^{-1}$ |
| Less than once a week | $-2.89 \times 10^{-1}$ | $-2.32 \times 10^{-1}$ | $-2.11 \times 10^{-1}$ |
| Number of household members | $-7.00 \times 10^{-1}$ | $-1.64 \times 10^{-2}$ | $5.27 \times 10^{-3}$ |
| Number of births | 1.01 | $-1.01 \times 10^{-1}$ | $1.18 \times 10^{-1}$ |
| **Contraceptive use** | | | |
| No method | – | – | – |
| Folkloric method | $1.90 \times 10^{-1}$ | $5.29 \times 10^{-2}$ | $-1.91 \times 10^{-1}$ |
| Traditional method | $3.24 \times 10^{-1}$ | $1.30 \times 10^{-1}$ | $-1.80 \times 10^{-1}$ |
| Modern method | $7.53 \times 10^{-1}$ | $2.64 \times 10^{-1}$ * | $-2.14 \times 10^{-1}$ |
| Constant | 1.20 | – | – |
| Total percentage points: | 3.28 ** | 0.652 | $-1.04$ |
| Percentage of overall prevalence difference: | 113.3% | 22.5% | $-35.8\%$ |

Statistical significance was assessed using $p$-values from two-sided Wald tests.
Previously married includes widowed, divorced, or separated.
Significance levels: *$p < 0.05$; **$p < 0.01$; ***$p < 0.001$.

proportion of secondary and higher education levels had a positive contribution to the increase in overweight and obesity prevalence over the last two decades (secondary: 1.07% increase, $p < 0.001$; higher: 0.664% increase, $p < 0.01$). This evidence is further supported by the significant endowment effect of wealth's richest quintile (0.958% decrease, $p < 0.01$), where it shows that the reduced proportion of women in the richest quintile resulted in a negative contribution to the rise in overweight and obesity prevalence. The overall coefficient effect, however, is the dominant driver of the increase in overweight and obesity prevalence from 2003 to 2022, accounting for 93.2% ($p < 0.001$) of the rise. Within the coefficient effect component, age, marital status, and region are the only significant variables.

## Discussion

This study examines the factors contributing to changes in overweight and obesity prevalence among women of reproductive age in Ghana using data from four GDHS survey years (2003, 2008, 2014, and 2022). The findings reveal that both overweight and obesity have increased significantly over the past two decades (2003–2022), with notable shifts in how individual characteristics influence these outcomes. These results have important implications for public health policy, highlighting the need for targeted interventions that address the evolving nature of these drivers.

Key drivers of overweight and obesity identified were age, marital status, education, wealth, residence, and lifestyle behaviours (e.g., television watching and contraceptive use). These patterns can be partly explained by cultural and societal perceptions that associate larger body size with wealth, health, and social prestige. In Ghana, as in many parts of sub-Saharan Africa, overweight and obesity have traditionally symbolised affluence, respectability, and success, particularly among women[32]. Higher education often coincides with greater socioeconomic status, which can reinforce such cultural ideals and influence lifestyle choices that contribute to weight gain[33,34].

Educational attainment is also frequently linked with urban residence and sedentary employment, characterised by desk-based work, reduced physical activity, and greater reliance on convenience foods[35]. In addition, women with higher education and income often have greater access to energy-dense, processed foods, while demanding work schedules may limit opportunities for meal preparation and regular exercise[13,35–37]. These factors combine to increase obesity risk, despite education's potential to promote healthier choices. Interestingly, these findings contrast with evidence from high-income countries, where higher education is often protective against overweight and obesity[38–41]. However, even in such contexts, sedentary employment and easy access to calorie-dense foods can offset education's protective effects. Public health campaigns in Ghana could therefore harness the positive aspects of education, such as enhanced knowledge of nutrition and physical activity, while addressing its unintended consequences.

Taken together, the interplay between lifestyle, socioeconomic status, and cultural norms that valorise a fuller body shape may help explain the rising prevalence of obesity among women with higher educational attainment in Ghana[34]. It is also important to note that cultural ideals of beauty and wellness often differ by gender. While men may be encouraged to demonstrate strength and wealth through body size, Ghanaian women, especially those with higher education, may be more exposed to competing ideals shaped by globalisation and media that promote thinness[34]. Nonetheless, the persistence of local norms associating overweight with wellness and prosperity may continue to influence dietary behaviours and weight outcomes[42].

The decomposition analysis further identifies structural changes in society beyond the individual behavioural factors, as drivers of the rising prevalence of overweight and obesity in Ghanaian women. For example, the reduced protective effect of being unmarried against weight gain may reflect shifting cultural norms or economic pressures that influence health and dietary behaviours among unmarried women[43,44]. Overall, these findings

**Table 3 | Multivariate decomposition of changes in overweight and obesity prevalence between Ghana DHS 2003 and 2022 (sensitivity analysis)**

| Variable | 2003–2022 overweight and obesity prevalence difference due to | | |
|---|---|---|---|
| | **Coefficients** | **Endowments** | **Interaction** |
| Age (years) | 12.4** | $9.40 \times 10^{-1}$** | $-2.54 \times 10^{-1}$ * |
| Marital status | | | |
| Married/Cohabiting | – | – | – |
| Never married | 4.95 *** | $-5.31 \times 10^{-1}$ ** | $-8.83 \times 10^{-1}$ *** |
| Previously married | $8.53 \times 10^{-1}$ * | $4.02 \times 10^{-2}$ | $-7.78 \times 10^{-2}$ |
| Educational attainment | | | |
| No education | – | – | – |
| Primary | $6.34 \times 10^{-1}$ | $-7.38 \times 10^{-1}$ *** | $2.87 \times 10^{-1}$ |
| Secondary | 2.87 | 1.07 *** | $-4.88 \times 10^{-1}$ |
| Higher | $-2.17 \times 10^{-1}$ | $6.64 \times 10^{-1}$ ** | $1.52 \times 10^{-31}$ |
| Wealth quintile | | | |
| Poorest | – | – | – |
| Poorer | $8.17 \times 10^{-1}$ | $9.49 \times 10^{-2}$ | $-4.67 \times 10^{-2}$ |
| Middle | $7.98 \times 10^{-1}$ | $2.76 \times 10^{-1}$ | $-7.03 \times 10^{-2}$ |
| Richer | $-2.61 \times 10^{-1}$ | $3.14 \times 10^{-1}$ | $1.54 \times 10^{-2}$ |
| Richest | $-1.08$ | $-9.58 \times 10^{-1}$ ** | $-1.52 \times 10^{-1}$ |
| Residence location | | | |
| Rural | – | – | – |
| Urban | 1.81 | $6.08 \times 10^{-1}$ *** | $-2.71 \times 10^{-1}$ |
| Region | | | |
| Ashanti | – | – | – |
| Brong-Ahafo | $-2.25 \times 10^{-1}$ | $-7.53 \times 10^{-3}$ | $3.11 \times 10^{-3}$ |
| Central | $-1.03$ * | $-1.54 \times 10^{-1}$ | $3.52 \times 10^{-1}$ * |
| Eastern | $-3.58 \times 10^{-1}$ | $9.38 \times 10^{-2}$ | $-1.19 \times 10^{-1}$ |
| Greater Accra | $-2.32$ *** | $6.56 \times 10^{-1}$ | $-1.42 \times 10^{-1}$ |
| Northern | $3.74 \times 10^{-1}$ | $-1.73 \times 10^{-1}$ * | $-5.55 \times 10^{-2}$ |
| Upper East | $2.96 \times 10^{-1}$ | $-4.34 \times 10^{-1}$ *** | $-1.65 \times 10^{-1}$ |
| Upper West | $2.88 \times 10^{-2}$ | $3.55 \times 10^{-1}$ *** | $2.52 \times 10^{-2}$ |
| Volta | $-3.64 \times 10^{-1}$ | $1.75 \times 10^{-1}$ * | $-1.14 \times 10^{-1}$ |
| Western | $-7.23 \times 10^{-1}$ * | $7.61 \times 10^{-2}$ | $-1.14 \times 10^{-1}$ |
| TV watching frequency | | | |
| Not at all | – | – | – |
| At least once a week | $5.80 \times 10^{-1}$ | 1.10 ** | $-1.57 \times 10^{-1}$ |
| Less than once a week | $5.30 \times 10^{-1}$ | $-2.19 \times 10^{-2}$ | $1.84 \times 10^{-2}$ |
| Number of household members | $-2.22$ | $4.98 \times 10^{-1}$ ** | $-2.57 \times 10^{-1}$ |
| Number of births | 1.47 | $-2.47 \times 10^{-1}$ * | $2.12 \times 10^{-1}$ |
| Contraceptive use | | | |
| No method | – | – | – |
| Folkloric method | $1.09 \times 10^{-1}$ | $6.52 \times 10^{-2}$ | $-6.84 \times 10^{-2}$ |
| Traditional method | $5.14 \times 10^{-1}$ | $1.55 \times 10^{-1}$ | $-1.72 \times 10^{-1}$ |
| Modern method | $6.27 \times 10^{-1}$ | $6.52 \times 10^{-1}$ *** | $-2.23 \times 10^{-1}$ |
| Constant | $-4.38$ | – | – |
| Total percentage points: | 16.5*** | 3.98*** | 2.76** |
| Percentage of overall prevalence difference: | 93.2% | 22.5% | $-15.6\%$ |

Previously married includes widowed, divorced, or separated.
Statistical significance was assessed using *p*-values from two-sided Wald tests.
Significance levels: *$p < 0.05$; **$p < 0.01$; ***$p < 0.001$.

highlight the dynamic nature of the factors influencing overweight and obesity, emphasising the need to address the population characteristics and their changing impacts to effectively manage these health concerns.

Age was identified as a significant predictor of overweight and obesity across all survey years. This finding is consistent with previous studies reporting that prevalence of overweight and obesity increases with age[45], suggesting that weight gain continues to be a cumulative risk in women in this age range[46]. This may be attributed to physiological changes in older women, such as slower metabolism, hormonal shifts, and reduced physical activity, which makes them more prone to weight gain[47,48]. While the decomposition analysis shows that age is a crucial factor, its impact on overweight and obesity has shifted over time, as evidenced by its significant contribution to the increase in prevalence between 2003 and 2022 through its coefficient effect. This coefficient effect may be due to changes over time in lifestyle, societal norms, or environmental factors, which have increased the impact of age on weight gain among women aged 15–49 years[49].

Our findings also highlight disparities by place of residence, with women in urban areas having significantly higher odds of obesity. This pattern reflects Ghana's ongoing urbanisation and the broader influence of urban environments, which are often associated with greater availability of processed foods and more sedentary lifestyles[50,51]. Similar trends have been documented in other developing countries, whereas in many high-income settings the relationship between urbanisation and overweight is reversed[52,53].

In Ghana, this "urban effect" may be explained by increased lifestyle choices, diverse food environments, and socioeconomic opportunities that accompany urban living. While urbanisation provides economic advantages, it also heightens exposure to obesogenic environments[54,55]. Public health initiatives could therefore prioritise promoting physical activity and healthier dietary choices among women in urban settings to help mitigate the rising prevalence of obesity. Additionally, the observed association between television watching and overweight highlights the importance of addressing sedentary behaviour within public health strategies.

As a counterpoint, rural habitation may offer certain protective factors. Rural communities are often characterised by physically demanding work, such as farming and other forms of manual labour, which contribute to higher daily energy expenditure. In addition, diets in rural settings typically rely more on locally produced, less processed foods, such as fruits, vegetables, and traditional staples, that are generally lower in fat, sugar, and salt compared to the energy-dense and ultra-processed foods more common in urban areas[56]. These patterns may partly explain the lower prevalence of overweight and obesity observed in rural populations[57]. Nonetheless, rural communities face their own nutrition-related challenges, including higher risks of undernutrition, food insecurity, and limited access to healthcare services[58]. Recognising both the protective and adverse aspects of rural and urban environments provides a more balanced understanding of how residence influences obesity risk in Ghana.

One notable finding is the role of modern contraceptive use, which is significantly associated with an increased risk of overweight, as has previously been suggested[23,24]. This association may be linked to hormonal influences of contraceptives on weight gain, although behavioural and social factors among contraceptive users could rather play a role since there is insufficient evidence linking contraceptives to weight change[59]. It is important to clarify that, although DHS collects information on specific contraceptive methods, our analysis used the aggregated measure of 'any modern method', which does not distinguish between hormonal and non-hormonal types.

Additionally, this association differed by setting: in urban areas, it was driven by modern contraceptive methods, while in rural areas, traditional methods such as period abstinence and withdrawal were significantly associated with overweight and obesity. Further research is needed to explore the underlying mechanisms behind these patterns. In terms of

policy implications, while contraceptive use itself should not be discouraged, it is important to address the potential link between certain contraceptive methods and weight gain through integrated reproductive health and nutrition counselling. For example, family planning services could include routine weight monitoring and tailored lifestyle advice on diet and physical activity for women using modern contraceptives. Such an approach would ensure that women receive comprehensive support to manage potential side effects while continuing to benefit from safe and effective contraceptive options.

Overall, this study shows that understanding the impact of drivers like age, marital status, education, and wealth on overweight and obesity prevalence is crucial. The decomposition analysis reveals that although changing population characteristics contribute to the prevalence of overweight and obesity, the rising trend is primarily driven by the evolving influence of these characteristics, reflected in shifts in their coefficient effects over time. These patterns emphasise the need to address contextual and lifestyle factors unique to these populations, underscoring the importance of dynamic, context-specific interventions that adapt to the changing drivers of overweight and obesity.

This study has some limitations that should be acknowledged. The analysis is based on cross-sectional survey data, which limits the ability to establish causal relationships between the identified risk factors and overweight or obesity[60]. Additionally, while the decomposition analysis provides insights into how changes in socioeconomic, behavioural, and demographic factors contribute to trends in obesity and overweight, it does not fully capture the potential interactions between individual and environmental or sociocultural factors over time. Future research should explore the causal pathways and mechanisms underlying these changes. Furthermore, this study focuses on a set of covariates that were available in the GDHS data, excluding other potentially influential factors such as dietary patterns or genetic predispositions[61]. Subsequent research and survey data collection could incorporate these variables to provide a more comprehensive understanding of the determinants of overweight and obesity in Ghana. Other research approaches, such as qualitative methods, could also provide a deeper, context-specific understanding of these drivers. Finally, a limitation worth highlighting is the lack of assessment of the food environment, which is recognised as a key determinant of dietary behaviours and a significant driver of overweight and obesity[62]. Future studies should incorporate measures of the food environment to provide a more comprehensive understanding of the contextual factors influencing obesity risk.

This study has several strengths, however, since it utilises data from nationally representative surveys spanning nearly two decades, which provides a comprehensive view of the trends in overweight and obesity among women of childbearing age in Ghana. The application of both multivariate logistic regression and Blinder–Oaxaca decomposition analyses allowed for a detailed examination of not just the prevalence of overweight and obesity, but also quantified the contributing factors driving the trend. By exploring the effects of various demographic, socioeconomic, and lifestyle factors, the study offers valuable insights into the dynamic nature of overweight and obesity risk, informing targeted public health interventions.

The findings of this study, while specific to women in Ghana, have implications that may extend to similar settings in sub-Saharan Africa and other low- and middle-income countries experiencing rapid urbanisation, economic growth, and shifting lifestyle patterns. The increase in overweight and obesity prevalence observed in this study reflects broader global trends, suggesting the drivers identified, such as age, education, wealth, and urban residence, are likely to be relevant in other contexts undergoing similar transitions. However, cultural, dietary, and regional differences may influence the specific nature and magnitude of these factors' effects[63]. Thus, while the results provide a valuable framework for understanding obesity dynamics in comparable environments, careful adaptation to local contexts is necessary when applying these insights to other populations.

The findings of this study also have important implications for public health policy in Ghana. To address the rising prevalence of overweight and obesity, interventions should target the changing nature of risk factors such as ageing, higher educational status, wealth, and marital status. Policies and public health efforts to mitigate the influence of high educational status and wealth on obesity should include strategies to encourage healthy dietary choices and physical activity, especially in urban areas. Additionally, addressing sedentary behaviour, such as excessive television watching, and understanding the effects of modern contraceptive use on weight are critical. A multi-faceted approach that considers the evolving socioeconomic and cultural dynamics is essential for creating effective, context-specific interventions to curb the obesity epidemic.

## Conclusions
This study underscores the complex and dynamic nature of the factors shaping overweight and obesity among women of childbearing age in Ghana. Beyond population characteristics, changing social and economic influences continue to redefine how risk factors operate over time. These findings highlight the need for policies and interventions that are responsive to shifting demographic patterns and evolving determinants, particularly those linked to education, wealth, and marital transitions. A nuanced and adaptive approach will be essential for effectively managing the growing burden of overweight and obesity in this population.

## Data availability
The data used in this study can be obtained for academic research purposes from the Demographic and Health Surveys (DHS) Program website (https://dhsprogram.com) by registering for access.

## Code availability
All R and Stata scripts used for data processing and statistical analyses are publicly available on Zenodo (https://doi.org/10.5281/zenodo.17936644)[31].

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

## Acknowledgements

Joseph P Mensah is funded by the University of Sheffield. This research was funded in part by the Wellcome Trust [Grant number 218462/Z/19/Z] and partly by the University of Sheffield Global Engagement Partnership Development Scheme. We gratefully acknowledge the University's support in fostering our collaboration with the University of Ghana, which has resulted in this initial output. For the purpose of open access, the author has applied a CC BY public copyright licence to any Author Accepted Manuscript version arising from this submission.

## Author contributions

Richmond Aryeetey (RAr) and Robert Akparibo (RA) conceptualised the study. J.P.M. and E.A. accessed the data. J.P.M. performed the data analysis and drafted the manuscript. RAr, RA, A.T., L.A.G., I.B., and M.B.K. contributed to the drafting and revising of the manuscript. All authors read and approved the final manuscript.

## Competing interests

The authors declare no competing interests.
