## [Transparent Peer Review file · Communications Medicine]

A multivariate decomposition analysis of drivers of overweight and obesity among Ghanaian women

Corresponding Author: Mr Joseph Mensah

Version 0:

Reviewer comments:

Reviewer #1

(Remarks to the Author)

This manuscript examines the drivers of overweight and obesity among Ghanaian reproductive-aged women using GDHS data from 2002, 2008, 2014, and 2022. It makes an important contribution to understanding these drivers, confirming findings from previous small-scale studies conducted in specific locations. While the identified drivers are not novel, the study highlights their universality across Ghana and their potential applicability to countries with similar population characteristics.

To enhance readability and clarity, I offer the following comments for the authors' consideration

Data analysis

In the data analysis section, authors should justify why descriptive statistical analyses were conducted in R while decomposition analyses were performed in Stata.

Results

In the results section, could the authors provide frequencies for all reported prevalences? For example, what was the frequency of individuals with no formal education in 2002, 2008, 2014, and 2022? This will help readers assess whether observed increases or decreases also reflect absolute changes. Additionally, when reporting aOR and CIs, please include p-values to indicate statistical significance.

Can the authors conduct a sub-analysis by residential status to determine if factors like wealth, education, and marital status persist among rural dwellers?"

Discussion

The probable explanation given to the rise in obesity with higher education could consider talking about the cultural context and perception within the Ghanaian society in which wellness is associated with overweight.

Limitation

Another limitation worth considering is the lack of assessment of the food environment which is also a key driver of overweight or obesity.

Conclusion

The conclusion should be revised to avoid repeating the results. For instance, using 'for example' suggests a rehash of findings rather than summarizing key insights. Instead, the conclusion should consist of declarative statements directly addressing the study objectives.

Reviewer #2

(Remarks to the Author)

The paper highlights important trends in overweight and obesity amongst reproductive age women in Ghana, from the

longitudinal Ghana Demographic and Health Surveys (GDHS) spanning 20 years. This is an important review for public health policy using verified local data.

Obesity is a current global pandemic and the effects on health are well documented, thus identifying the contributing factors and trends from local data is commendable.

Background: Lines 77-79 compare data in different age groups: 20-49 and 15-49 years; comparing similar age groups would be more representative.

Lines 86-88: The 3 references quoted (16,17,18) for drivers of obesity in Ghana are not pertinent to Ghana.

Methods:

The methodology and statistical analysis could be better explained, and would benefit from expert statistical review.

Discussion: The paper references previous studies on the subject matter done in Ghana but does not elaborate on their findings and what their study adds.

Key drivers of obesity and overweight and their changing trends over the years are identified- age, education, marital status, urbanisation, contraception usage- and some explanation given for some.

Urbanisation is shown to be associated with higher odds of obesity, the authors could have mentioned some beneficial effects of rural habitation as a counterpoint.

'Modern' contraceptive use is highlighted as another association, with the focus on hormonal contraception with no evidence of the type of contraception referred to in the base data.

No objective policy solutions to correct the identified trends are offered.

Reviewer #3

(Remarks to the Author)

In this study, the authors sought to examine the drivers of overweight and obesity among women in Ghana using pooled data from the 2003 to 2022 Ghana Demographic and Health Surveys (GDHS). The manuscript presents a relevant and timely investigation into a pressing public health issue. The discussion of the rationale for the study is informative, and the data sources and methods are generally well described. The analysis appears to be thoughtfully conducted, and the results are discussed in a meaningful way. However, several areas in the manuscript require revision to enhance clarity, strengthen methodological rigour, and better contextualise the findings. Please see my detailed comments below:

Introduction – Given the study's focus on sociodemographic and economic factors influencing overweight and obesity, the introduction should be more explicitly grounded in these themes. At present, the discussion centres on general risk factors for obesity, which limits the specificity and relevance to the research aims. The introduction would benefit from a clearer articulation of the study's contribution to the literature, particularly regarding how it addresses gaps in existing research on socioeconomic and demographic disparities in obesity prevalence among Ghanaian women.

Analysis – Considering the stratified sampling design of the GDHS, it is important that the analysis accounts for the complex survey design, particularly potential clustering at the household or enumeration area level. This aspect is currently not discussed. I recommend that the authors consider using a hierarchical or multilevel mixed-effects logistic regression model to account for such clustering. Alternatively, the use of the `vce(cluster ...)` or `svy` commands in Stata would be appropriate methods for obtaining robust standard errors and accounting for survey design effects.

Discussion – The discussion would benefit from a more in-depth examination of regional patterns and temporal trends in overweight and obesity prevalence, especially given the inclusion of data from multiple survey waves and the availability of region-level variables. It would be helpful to explore whether observable changes in prevalence have occurred over the years and what contextual or policy-related factors might explain these trends. This would add depth to the discussion and strengthen the relevance of the findings for informing interventions and public health planning.

Version 1:

Reviewer comments:

Reviewer #2

(Remarks to the Author)

The authors should be commended for taking on board the initial reviewer recommendations and amending their paper accordingly.

The revised manuscript provides a detailed review of the temporal trends and prevalence of overweight and obesity in reproductive aged women in Ghana from four GDHS surveys over 2 decades, with their decomposition analysis providing a detailed assessment of the main drivers.

Though not novel, this review from a large representative and validated national dataset provides ample evidence and makes some recommendations for public health policy decisions and further research in the ongoing fight against obesity as a global pandemic.

Consider making a grammatical change in the sentence below with the inserts-
In 2003, 25.3% of women were classified as having (being) overweight or obesity (obese). This percentage rose to 29.9% in 2008, 40.1% in 2014, and 43.0% in 2022. This represents a 17.7 percentage point increase over nearly two decades.

Reviewer #3

(Remarks to the Author)

All my prior comments have been addressed

Regarding the response to the other reviewer: I've now reviewed the manuscript, and it appears that the authors have addressed all concerns raised in the previous review. The only remaining issue pertains to the frequency distribution. While the authors provide a valid rationale for not including the specific frequencies requested by the reviewer, I recommend that they consider one of the following options:

1. Include both the weighted and unweighted prevalence in the table, or
2. Include only the weighted prevalence but add a footnote indicating that the frequencies represent unweighted (raw) counts from the study sample.

I agree with the previous reviewer on the importance of presenting unweighted frequencies for clarity and transparency.

Response to Reviewers – Communications Medicine

Reviewer's Comment	Response from Authors
Referee #1: Africa, epidemiology expertise	
This manuscript examines the drivers of overweight and obesity among Ghanaian reproductive-aged women using GDHS data from 2002, 2008, 2014, and 2022. It makes an important contribution to understanding these drivers, confirming findings from previous small-scale studies conducted in specific locations. While the identified drivers are not novel, the study highlights their universality across Ghana and their potential applicability to countries with similar population characteristics.	
To enhance readability and clarity, I offer the following comments for the authors' consideration	
Data analysis In the data analysis section, authors should justify why descriptive statistical analyses were conducted in R while decomposition analyses were performed in Stata.	Thank you for these helpful comments. We chose to conduct different components of the analysis using different software platforms based on their specific strengths and suitability for the analytical tasks involved. Descriptive statistical analyses were performed in R, which offers robust and flexible tools for data manipulation, visualisation, and summary statistics, making it particularly effective for exploratory data analysis. Decomposition analyses, on the other hand, were conducted in Stata due to its well-established capabilities for implementing non-linear decomposition methods, including logistic regression models. In particular, commands such as <code>mvdcmp</code> in Stata are specifically designed to handle non-linear models and allowed us to decompose differences across groups in a methodologically appropriate manner. By using both R and Stata, we were able to take advantage of the unique analytical capabilities of each software to ensure rigorous and efficient analysis.
Results In the results section, could the authors provide frequencies for all reported prevalences? For example, what was the frequency of individuals with no formal education in 2002, 2008, 2014, and 2022? This will help readers assess whether observed increases or decreases also reflect	Thank you again for the helpful comment. The reported prevalence estimates are survey-weighted, reflecting the weighted proportion of individuals with overweight/obesity in the population, and thus reporting the unweighted frequencies, using the raw count from the study sample as suggested, may be misleading without

absolute changes. Additionally, when reporting aOR and CIs, please include p-values to indicate statistical significance.	proper context. We therefore kindly request that the reviewer reconsider this suggestion and allow us to retain the data as presented. We have however, added p-values, as suggested, to complement the reported adjusted odds ratios and confidence intervals, providing additional statistical context for the observed associations.
Can the authors conduct a sub-analysis by residential status to determine if factors like wealth, education, and marital status persist among rural dwellers?"	Thank you also for your suggestion to do a sub-analysis by residential status. We have now made this additional analysis, and the findings have been reported and discussed. See the results table included as supplementary material.
Discussion The probable explanation given to the rise in obesity with higher education could consider talking about the cultural context and perception within the Ghanaian society in which wellness is associated with overweight.	Thank you for this helpful suggestion. We have added a few paragraphs in the discussion section to address this important point, which we agree is essential.
Limitation Another limitation worth considering is the lack of assessment of the food environment which is also a key driver of overweight or obesity.	Well spotted, thank you. We have incorporated a line in the limitations section to address this important suggestion. We have added this statement to the limitation section "a limitation worth highlighting is the lack of assessment of the food environment, which is recognised as a key determinant of dietary behaviours and a significant driver of overweight and obesity. Future studies should incorporate measures of the food environment to provide a more comprehensive understanding of the contextual factors influencing obesity risk"(page 11)
Conclusion The conclusion should be revised to avoid repeating the results. For instance, using 'for example' suggests a rehash of findings rather than summarizing key insights. Instead, the conclusion should consist of declarative statements directly addressing the study objectives.	Thank you. We have revised the conclusion for clarity and to avoid repetition of the results.
Referee #2: Africa, public health expertise	

The paper highlights important trends in overweight and obesity amongst reproductive age women in Ghana, from the longitudinal Ghana Demographic and Health Surveys (GDHS) spanning 20 years. This is an important review for public health policy using verified local data. Obesity is a current global pandemic and the effects on health are well documented, thus identifying the contributing factors and trends from local data is commendable.	
Background: Lines 77-79 compare data in different age groups: 20-49 and 15-49 years; comparing similar age groups would be more representative. Lines 86-88: The 3 references quoted (16,17,18) for drivers of obesity in Ghana are not pertinent to Ghana.	We thank the reviewer for this observation. We have revised the background section to ensure age groups are presented consistently and now use comparable categories. In addition, we have replaced the cited references with more relevant studies that directly address the drivers of obesity in Ghana.
Methods: The methodology and statistical analysis could be better explained and would benefit from expert statistical review.	We have provided further explanation to the analysis based on the other reviewers' helpful suggestion. We hope this helps address your concerns. It is worth saying that three of the authors are very experienced statisticians and they have all revisited the data and reviewed the report to ensure accuracy.
Discussion: The paper references previous studies on the subject matter done in Ghana but does not elaborate on their findings and what their study adds. Key drivers of obesity and overweight and their changing trends over the years are identified- age, education, marital status, urbanisation, contraception usage- and some explanation given for some. Urbanisation is shown to be associated with higher odds of obesity, the authors could have mentioned some beneficial effects of rural habitation as a counterpoint. 'Modern' contraceptive use is highlighted as another association, with the focus on hormonal contraception with no evidence of the type of contraception referred to in the base data. No objective policy solutions to correct the identified trends are offered.	We have expanded the discussion to be more explicit and have highlighted the beneficial effect of rural habitation as a counter point. In terms of the comment related to modern contraceptive, we thank the reviewer for raising this important point. While the DHS does provide information on specific contraceptive methods, our analysis relied on the aggregated measure of any modern method, which does not distinguish between hormonal and non-hormonal types. We have clarified this limitation in the manuscript to avoid over-interpretation of the findings (page 11).

Referee #3: Global public health expertise

In this study, the authors sought to examine the drivers of overweight and obesity among women in Ghana using pooled data from the 2003 to 2022 Ghana Demographic and Health Surveys (GDHS). The manuscript presents a relevant and timely investigation into a pressing public health issue. The discussion of the rationale for the study is informative, and the data sources and methods are generally well described. The analysis appears to be thoughtfully conducted, and the results are discussed in a meaningful way. However, several areas in the manuscript require revision to enhance clarity, strengthen methodological rigour, and better contextualise the findings. Please see my detailed comments below:

Introduction – Given the study's focus on sociodemographic and economic factors influencing overweight and obesity, the introduction should be more explicitly grounded in these themes. At present, the discussion centres on general risk factors for obesity, which limits the specificity and relevance to the research aims. The introduction would benefit from a clearer articulation of the study's contribution to the literature, particularly regarding how it addresses gaps in existing research on socioeconomic and demographic disparities in obesity prevalence among Ghanaian women.

Thank you. We have revised the background to highlight the study contribution to the literature.

Analysis – Considering the stratified sampling design of the GDHS, it is important that the analysis accounts for the complex survey design, particularly potential clustering at the household or enumeration area level. This aspect is currently not discussed. I recommend that the authors consider using a hierarchical or multilevel mixed-effects logistic regression model to account for such clustering. Alternatively, the use of the `vce(cluster ...)` or `svy` commands in Stata would be appropriate methods for obtaining robust standard errors and accounting for survey design effects.

Thank for this comment. We have indicated that survey-weighted logistic regression models, using the survey package in R, was applied to account for the complex survey design. This package, as well as the `svy` commands in Stata, were used, and this has been further described in the methods.

Discussion – The discussion would benefit from a more in-depth examination of regional patterns and temporal trends in overweight and obesity prevalence, especially given the inclusion of data from multiple survey waves and the availability of region-level variables. It would be

Thank you. We have expanded the discussion to reflect this comment. We appreciate.

helpful to explore whether observable changes in prevalence have occurred over the years and what contextual or policy-related factors might explain these trends. This would add depth to the discussion and strengthen the relevance of the findings for informing interventions and public health planning.	
---	--

Response to Reviewers – Communications Medicine

Reviewer's Comment	Response from Authors
Reviewer #2 (Remarks to the Author):	
The authors should be commended for taking on board the initial reviewer recommendations and amending their paper accordingly. The revised manuscript provides a detailed review of the temporal trends and prevalence of overweight and obesity in reproductive aged women in Ghana from four GDHS surveys over 2 decades, with their decomposition analysis providing a detailed assessment of the main drivers. Though not novel, this review from a large representative and validated national dataset provides ample evidence and makes some recommendations for public health policy decisions and further research in the ongoing fight against obesity as a global pandemic.	
Consider making a grammatical change in the sentence below with the inserts- In 2003, 25.3% of women were classified as having (being) overweight or obesity (obese). This percentage rose to 29.9% in 2008, 40.1% in 2014, and 43.0% in 2022. This represents a 17.7 percentage point increase over nearly two decades.	Thank you for the remarks and the grammatical suggestion. We have made the suggested changes to improve the clarity of the sentence.
Reviewer #3 (Remarks to the Author):	
All my prior comments have been addressed Regarding the response to the other reviewer: I've now reviewed the manuscript, and it appears that the authors have addressed all concerns raised in the previous review. The only remaining issue pertains to the frequency distribution. While the authors provide a valid rationale for not including the specific frequencies requested by the reviewer, I recommend that they consider one of the following options:	
1. Include both the weighted and unweighted prevalence in the table, or 2. Include only the weighted prevalence but add a footnote indicating that the frequencies represent unweighted (raw) counts from the study sample. I agree with the previous reviewer on the importance of presenting unweighted frequencies for clarity and transparency.	We thank the reviewer for the helpful comment regarding frequency reporting. In line with option 2, we report only the weighted prevalence and have added a footnote to clarify this distinction.